



# Uncertainty of the hourly average concentration values derived from non-continuous measurements

László Haszpra, Ernő Prácser

Geodetic and Geophysical Institute, Research Centre for Astronomy and Earth Sciences, Sopron, H-9400, Hungary

*Correspondence to*: László Haszpra (haszpra.l@gmail.com)

**Abstract.** Continental greenhouse gas monitoring networks extensively use tall towers for higher spatial representativeness. In most cases, several intakes are built along the tower to give information also on the vertical concentration profile of the components considered. Typically, a single gas analyzer is used, and the intake points are sequentially connected to the instrument. It involves that the continuous concentration signal is only sampled for discrete short periods at each intake

points, which does not allow the perfect reconstruction of the original concentration variation. It increases the uncertainty of the calculated hourly averages usually used by the transport and budget models. The purpose of the study is to give the data users an impression on the potential magnitude of this kind of uncertainty, as well as how it depends on the number of intakes sampled, on the length of the sampling period at each intake, on the season, and on the time of the day. It presents how much improvement can be achieved using linear or spline interpolation between the measurement periods instead of the

simple arithmetic averaging of the available measurements. Although the results presented here may be site-specific, the study calls attention to the potentially rather heterogeneous spatial and temporal distribution of the uncertainty of the hourly average concentration values derived from tall-tower measurements applying sequential sampling.

## 1 Introduction

Continental greenhouse gas (GHG) monitoring networks extensively use tall towers for the measurements for obtaining

higher spatial representativeness (Tans, 1991; Bakwin et al., 1995; Wofsy and Harriss, 2002; Vermeulen, 2007; Gerbig et al., 2009; Kadygrov et al., 2015; Oney et al., 2015; White et al., 2019; ICOS RI, 2020). In most cases, taking advantage of the tower, several intakes are mounted along the tower to get information on the vertical distribution of the component considered. Typically, a single gas analyzer is used, and the intake lines are connected sequentially to the instrument (Bakwin et al., 1998; Haszpra et al., 2001; Vermeulen, 2007; Thompson et al., 2009; Popa et al., 2010; Winderlich et al.,

2010; Vermeulen et al., 2011; Andrews et al., 2014; Satar et al., 2016; Conil et al., 2019; ICOS RI, 2020). Using a single instrument is not only cheaper but it also avoids the scale differences among the instruments, which could inevitably occur even at frequent synchronization. In the case of continuous in situ measurements, international databases, like World Data Centre for Greenhouse Gases of World Meteorological Organization (https://gaw.kishou.go.jp/), ICOS Carbon Portal (https://www.icos-cp.eu/icos-carbon-portal) or ObsPack maintained by National Oceanic and Atmospheric Administration,



U.S.A. (https://www.esrl.noaa.gov/gmd/ccgg/obspack/), store and disseminate hourly average concentration values, and these values are used in the different mathematical models (atmospheric budget models, inverse transport models, etc. – see e.g. Ciais et al., 2010; Rivier et al., 2010; Bergamaschi et al., 2015; Diallo et al., 2017; Shirai et al., 2017; Bergamaschi et al., 2018; Lin et al., 2018; Chevallier et al., 2019). As long as an instrument receives air from a given intake continuously, the hourly average concentration is the temporal integral of the instrument's signal for the given hour. However, if the

instrument switches among the different intakes, only short samples of the continuous concentration signal are available from each intake for the estimation of the hourly average, which increases the uncertainty of the hourly average concentration values reported. According to the Nyquist–Shannon sampling theorem fluctuation having a higher frequency than half of the sampling one cannot be reconstructed from the data recorded. At continental carbon dioxide monitoring stations surrounded by vegetation, significant short-term concentration changes are rather common, especially during the

growing season due to the activity of the plants and the dynamic processes of the atmosphere. Losing the high-frequency part of the spectrum may introduce significant uncertainty into the hourly average concentration values especially in certain seasons of the year and in certain periods of the days, which adds to the common instrument noise and scale uncertainty. While the reduction of the instrument noise would require longer signal integration time, it would also result in the loss of high-frequency concentration fluctuation. Significant concentration changes, like those during the morning transition period,

when the low-level inversion breaks up, or during a frontal passage (Pal et al., 2020), cannot be followed properly by discrete sampling either.

For the reduction of the uncertainty derived from episodic sampling both physical and mathematical methods can be applied. The physical method involves a buffer volume in the sampling line, which physically integrates the high-frequency concentration fluctuation. A properly designed weighted averaging scheme can significantly reduce the uncertainty of the

hourly average values (Cescatti et al., 2016). However, in this case, it is difficult to determine the exact start of the integration period for which the calculated hourly average is characteristic, and spike detection (El Yazidi et al., 2018) cannot be applied for quality assurance.

In the present paper, we analyze the uncertainty of the hourly average carbon dioxide concentration values originating from the discrete episodic signal-sampling for the case when no buffer volume is applied in the measuring system. We use

measurement data from a mid-continental monitoring site to show how much uncertainty is introduced by the episodic sampling and how it varies in time.

## 2 Methodology and data

For the study, the continuous in situ carbon dioxide concentration measurements carried out at Hegyhátsál tall-tower GHG monitoring site (46°57'N, 16°39'E, 248 m above the sea level, NOAA/WMO GAW code: HUN) at 82 m above the ground

are used. The tower is located in a European mid-continental rural environment, at a low elevation above the sea level, in the temperate climate zone, and it is surrounded by natural and agricultural vegetation (Haszpra et al., 2001). At such a site, the





vegetation and the dynamic processes of the atmosphere may generate significant short-term changes in the concentration of carbon dioxide in the planetary boundary layer sampled by a tower. At Hegyhátsál tall-tower GHG monitoring site a Picarro G2301 CRDS analyzer (Picarro, Inc., Santa Clara, California, USA) is operated for carbon dioxide (and methane)

measurements. The instrument provides the dry mole fraction data, also referred to as concentration in this paper, at a 5-second temporal resolution. For the study, the data measured in 2018 were selected. This data series is considered continuous from which the *true* hourly average concentrations are calculated for each hour of the year. Different subsamples of this series were generated to simulate the multi-level measurements when the analyzer cycles through the monitoring elevations sampling each of them only for a short time.

In this study, we suppose 2, 3, 4, and 5 measurement elevations and study a wide range of sampling periods. The length of the sampling period was determined so that each measurement elevation is sampled equal times within the hour (Fig. 1). So, the maximum sampling period is 1800 s for 2 intakes, and it is 720 s for 5 intakes. In such extreme situations, only one sampling would be performed at each measurement elevation in each hour.

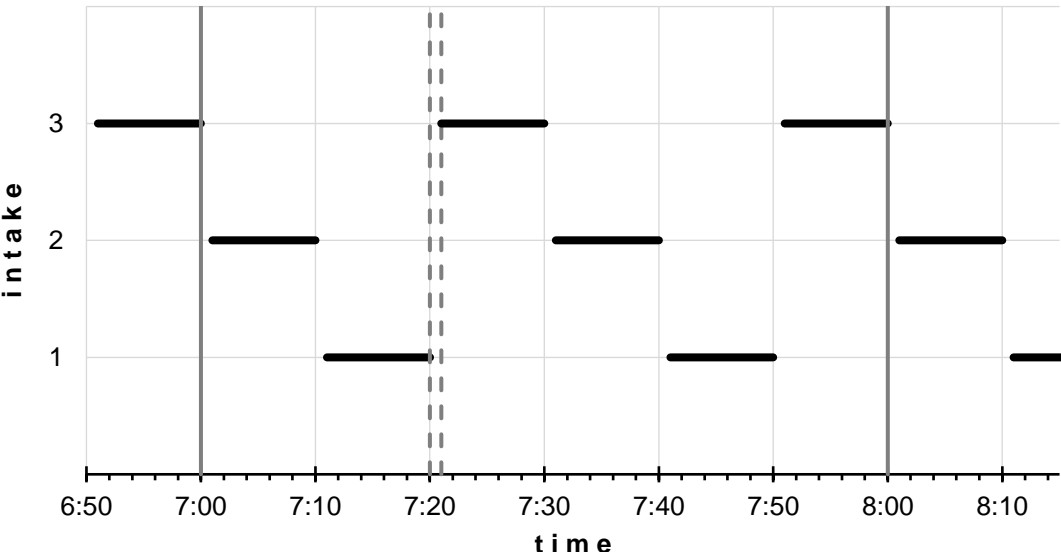

**Figure 1: Schematics of an example measurement protocol with 3 intakes and 600 s sampling time providing two complete cycles within an hour. Vertical dashed lines represent the 1-minute flush-time after switching the intakes and before the signal integration. The signal integration period is indicated by thick black line segments**

The minimum length of the sampling period is technically determined: after switching the intakes, a certain time is needed

for flushing the tubing and instrument before valid measurements can be performed. The length of this time depends on the design of the monitoring system. In our case, the intake lines from the tower are permanently ventilated, and only a relatively small volume (short tubes, selector valve, Nafion drier tube, etc. – see Haszpra et al. (2001) for general design) has to be





flushed at each switch before the air to be measured arrives at the measuring cell of the analyzer. Our system is operated at a flow rate of ~220 ml air per minute, and at this flow rate it takes <10 s for the sample air to reach the measurement cell of the analyzer after switching the intake. Theoretically, flushing the measurement cell follows an exponential model where the air from Intake 2 gradually replaces the air remaining in the measuring cell from Intake 1 at continuous mixing. Also theoretically, the perfect flushing needs infinite time in this way. In practice, the difference between the true concentration of air from Intake 2 and the concentration in the measuring cell quickly disappears in the range of the noise of the instrument. Our experience with standard gases shows that even if the initial concentration difference is as high as 70 $\mu$mol mol$^{-1}$, the deviation of the concentration in the measuring cell from the true value falls below 0.1 $\mu$mol mol$^{-1}$ within 35-45 s. Example test measurements are presented in Figure S1 in the Supplementary material.

Another question is the length of the signal integration after the acceptable flushing to reduce the effect of the instrument noise in the measured concentration. Laboratory tests with standard gases may help the determination of the optimum averaging time (Yver Kwok et al., 2015). However, laboratory tests do not help much if the concentration in the sample gas changes during the averaging period as it happens during real-world atmospheric measurements. Long averaging time involves less frequent measurement at each intake and increased uncertainty of the calculated hourly averages. On the other hand, at a too-short averaging time the instrument noise may dominate the result. For the determination of a reasonable trade-off between these conflicting requirements, a linear regression model was constructed. As long as the instrument noise dominates above the natural variability the slope of the linear regression line fitted to the data series does not differ statistically significantly from zero (null-hypothesis). We calculated the slopes of the linear regression lines as the function of the length of the involved data series. Using the F-test at a significance level of 5 % for the null-hypothesis mentioned above, it turned out that even a 40 s long data series show a statistically significant linear trend in as much as 38 % of the cases, which increases with the length of the data series involved. Taking into account the above experience we selected the shortest sampling time for this study as 100 s (120 s for 5 intakes) including 60 s flushing and 40 s (60 s for 5 intakes) signal averaging.

The dynamic processes in the planetary boundary layer (PBL) significantly differ from those above that. In the present study, we selected only those periods when the measurement elevation (82 m above the ground) was well within the planetary boundary layer, the depth of the PBL was at least 120 m at both the beginning and end of the hour considered. The PBL depth data were taken from the ERA5 reanalysis dataset of the European Centre for Medium-Range Weather Forecasts (Copernicus Climate Change Service, 2017) with 1 h temporal resolution. Using the criterion mentioned above, we could avoid the periods when the top of the PBL is around the measurement elevation, and its fluctuation causes extreme variability in the carbon dioxide concentration. These periods are also difficult to handle in the atmospheric models.

In addition to the arithmetic averaging of the data available in the given hour, more sophisticated methods were also applied for the estimation of the hourly averages. In this study, the linear interpolation and the cubic spline interpolation were tested on how much they can reduce the uncertainty of the hourly average concentrations relative to the common arithmetic averaging. The timestamps of the data for the linear and spline interpolations were the middle of the sampling periods, not





considering the flush time. For cubic spline interpolation, the SPLINE_P procedure of IDL 6.3 (Interactive Data Language, Harris Geospatial Solutions) was applied. Using the cubic spline interpolation the concentration was reconstructed at 5 s resolution and integrated through the given hour.

**3 Results and discussion**

Typically, the hourly average concentration is calculated as the arithmetic mean of the average concentrations of the short periods sampled. The resulted value randomly deviates from the true hourly average due to the unsampled periods. The probability distribution of the deviations is far from Gaussian as it can be seen in Figure 2. The rather peaked distribution can be approximated by Cauchy distribution fairly well. As it is reasonable to suppose that the mean deviation is zero, the density of the Cauchy distribution can be written in a simple form:

$$f(x, \gamma) = \frac{\gamma}{\pi(\gamma^2 + x^2)} \ ,$$

where $\gamma$ is the so-called scale parameter. It gives half of the interquartile range of the distribution. As the distribution is symmetric to zero, $\gamma$ also gives the median of the absolute values of the deviations. The probability distribution of the deviations has rather long tails. Therefore, for practical purposes, it is reasonable to know the probability of high deviations, which might significantly distort the results of atmospheric models. In this paper, the 90-percentile values are used to characterize the extreme values, which still have a non-negligible probability. Cauchy distribution was fitted to the empirical distribution using MPFITEXPR function written in IDL (Markwardt, 2009) applying Levenberg-Marquardt least-squares fit. With the increasing length of the sampling period, the uncertainty of the calculated hourly average is also increasing due to the fewer measurements per intake. The increased uncertainty is indicated by the increased $\gamma$ value. Figure 2 shows the probability distribution of the deviation from the true value for 3 intakes and two sampling periods, 120 s and 400 s. In the first case (120 s), a certain intake is sampled 10 times during an hour, while in the latter case (400 s) it is sampled only 3 times. The difference is remarkable. The full set of distributions can be found in the Supplementary material.

It is fairly obvious that there is little difference among the results of arithmetic averaging, linear interpolation, and spline interpolation if the sampling period is short, that is a given intake is sampled frequently, and only short periods are missed by the measurements. The advantage of the more sophisticated methods appears when the sampling period is longer and fewer samples are available within the hour. The linear interpolation can better estimate the concentration at the beginning and the end of the hour as it also uses measurements from the previous and subsequent hours. Cubic spline interpolation follows the temporal course of the concentration as much as it is possible at the limited number of data.



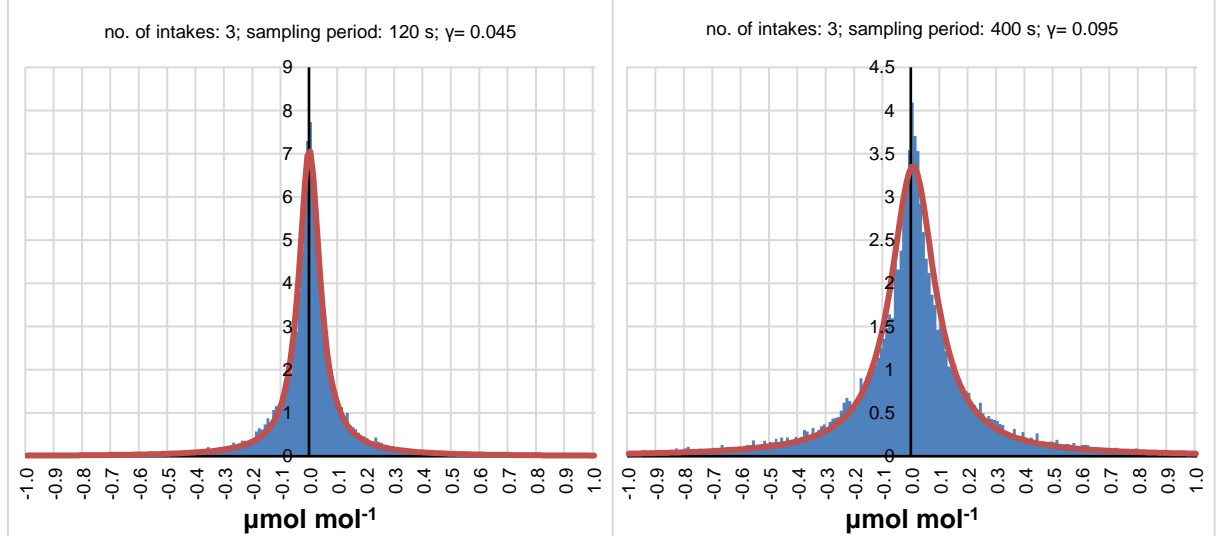

**Figure. 2: Distribution of the deviations of the hourly averages estimated as the arithmetic average of the available measurements from the true hourly averages for 3 intakes with 120 s (left panel) and 400 s (right panel) sampling periods. Scale parameters of the Cauchy distribution fitted are 0.045 and 0.095, respectively**

Figure 3 visualizes the difference between the methods and their results in an example of a typical summer morning. Figure 4 shows the γ values (equivalent with the half of the interquartile range or the median of the absolute deviations) and the 90-percentile values of the absolute deviations for the different number of intakes as a function of the sampling time and averaging method (arithmetic average, linear interpolation, cubic spline interpolation).

The activity of the vegetation, uptake and release of carbon dioxide, and the dynamics of the atmosphere dispersing it in the air have significant seasonal and diurnal variations, which also influence the uncertainty of the calculated hourly averages. To get an insight into the temporal variation of the uncertainty we grouped the calculated deviations from the true values by month and by the time of the day with hourly resolution. At such a resolution, the available data are not enough for the reliable estimation of the scale parameter of the Cauchy distribution. Instead, we calculated the approximate median and 90-percentile of the absolute deviations based on the empirical distributions. The results should be considered rather a qualitative than quantitative one but they indicate when the modelers should be aware of potentially high uncertainty of the data, and what the magnitude can be. Figure 5 shows a color-coded temporal distribution map of the rate of the uncertainty for 3 intakes and two sampling periods, 120 s and 600 s, giving 10 and 2 measurement cycles per hour, respectively. For other sampling periods and numbers of intakes, the data are presented in the Supplementary material.



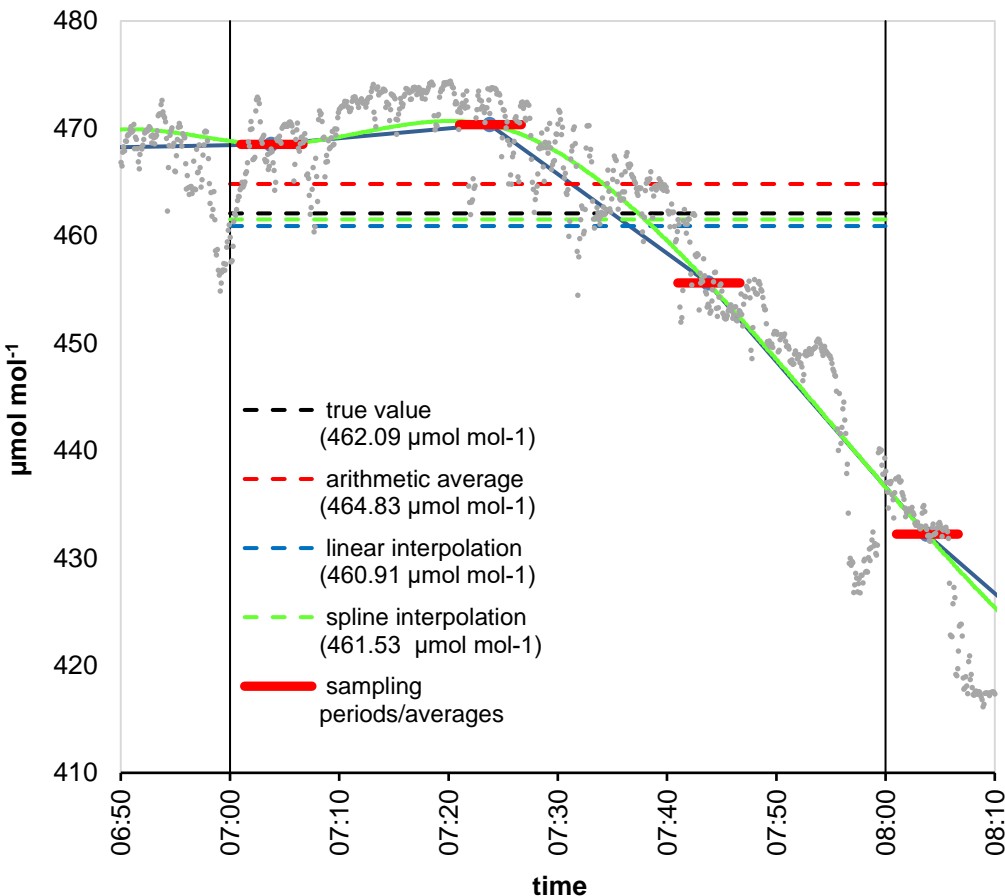

**Figure 3: Difference between the hourly averages estimated by arithmetic averaging, linear interpolation and spline interpolation, and the true value in a typical summer morning hour (2 May 2018). Gray dots are the measurement data with 5 s temporal**
**resolution, while the thick red lines give the sampling periods and their average concentrations supposing 3 intakes and 400 s sampling time (60 s flushing + 340 s signal integration at each intake). The height of the planetary boundary layer changed from 399 m to 555 m from 7:00 to 8:00**





**Figure 4: The median (p50) and the 90-percentile (p90) values of the absolute deviation from the true hourly average as a function of sampling time and number of intakes calculated from the fitted Cauchy-distribution**








**Figure 5: The temporal variation of the 90-percentile of the absolute deviations of the hourly averages calculated as the arithmetic average of the available measurements from the true values for 3 intakes, and for 120 s (upper panel) and 600 s (lower panel) sampling time**





Our analysis attributes quantitative results to the qualitatively more or less expectable findings. As it can be seen in Fig. 4,
the shorter the sampling period the lower the uncertainty of the calculated hourly averages, and the more sophisticated
methods result in reduced uncertainty. While both the systematic errors derived from e.g. scale bias and the random noise of
the gas analyzers are relatively stable within a short time, the uncertainty in the hourly averages caused by the discrete
sampling of the continuous signal shows a remarkable temporal variation (Fig. 5). It is lower during the afternoon hours
when the atmosphere is fairly well-mixed but may be huge during the morning transition periods due to the sudden
concentration changes in the planetary boundary layer monitored, which cannot be properly followed by episodic sampling.
Due to the seasonal variation in both the activity of the vegetation and the dynamics of the atmosphere, the uncertainties are
higher in summer than in winter.

In the case of arithmetic averaging, the mean of the absolute values of deviations from the true hourly averages remain
below 0.3 µmol mol$^{-1}$ during the winter (Dec-Feb) afternoon hours (the 90-percentile values are <0.8 µmol mol$^{-1}$) supposing
at least two measurement cycles per hour. It is somewhat higher in summer (Jun-Aug) but still less than 0.5 µmol mol$^{-1}$ (the
90-percentile values are <0.9 µmol mol$^{-1}$). Although the mixing of the atmosphere is more vigorous during summer
afternoons the intensive carbon dioxide uptake by the vegetation causes higher concentration fluctuation in the atmosphere
than it is experienced in winter.

The uncertainty of the hourly averages derived from episodic samplings is much higher during the morning transition period
when the concentration may change several tens of µmol mol$^{-1}$ within an hour. The uncertainty is especially high in summer
when the diurnal amplitude of the carbon dioxide concentration is the highest due to the intensive photosynthesis/respiration
of the vegetation.

If at least two measurement cycles per hour are assumed again, the mean uncertainty remains below 0.4 µmol mol$^{-1}$ during
the morning transition hours in winter (the 90-percentile values are <1.5 µmol mol$^{-1}$) but it may reach 1.7 µmol mol$^{-1}$ in
summer (the 90-percentile values may be as high as 5 µmol mol$^{-1}$). During these periods, the sampling frequency is critical.
The higher the sampling frequency the better followed the concentration course, and the lower the uncertainty of the
estimated hourly average. At three cycles per hour, the mean uncertainty does not exceed 1.2 µmol mol$^{-1}$ in the data series
studied (the 90-percentile values are <3 µmol mol$^{-1}$), which is further reduced to 0.9 µmol mol$^{-1}$ (the 90-percentile values are
<2.5 µmol mol$^{-1}$) when at least four cycles were assumed. The evening collapse of the convective boundary layer causes a
relatively fast increase in the concentration in the boundary layer but the rate of the change is lower than during the morning
build-up, therefore, the resulted uncertainty in the estimated hourly average concentrations is also lower.

When the sampling time is short, each intake is sampled relatively frequently, therefore the more sophisticated averaging
methods do not give significant improvements in the uncertainty of the calculated hourly averages. Their advantage appears
when the sampling frequency is low and/or the concentration changes are significant within the hour because the
interpolation methods follow the actual temporal course of the concentration better than the short term averages. Although
the temporal distribution of the uncertainty does not change, the high values (summer morning periods) are significantly
reduced as it can be seen in Fig. 6. Theoretically, the cubic spline interpolation follows the course of the concentration more





faithfully than the linear interpolation, however, a significant difference cannot be seen in their performance under the conditions studied here.

The full set of the uncertainty maps (different number of intakes, sampling periods, methods) can be found in the Supplementary material.

**Figure 6: The seasonal variation of the median (p50) and the 90-percentile (p90) values of the absolute deviations of the hourly averages calculated using the different methods (arithmetic averaging, linear interpolation, spline interpolation of the available data) in the case of 3 intakes, and for 120 s (upper panels) and 600 s (lower panels) sampling time. The left panels characterize the**
**morning transition periods (6-8 h), while the right panels show the corresponding values for the early afternoon hours (13-15 h)**





It should be emphasized that the numerical results presented here may be highly site-specific. They may depend on the height of the sampling elevation above the ground, on the geographical location and environment of the monitoring site, and partly on instrument setup. Sources and sinks of carbon dioxide are located at the surface, and the fluctuation generated by the surface processes gradually attenuates with height (Stuhl, 1999). The relative role of the high-frequency part of the

spectra is reduced at higher elevations, and so the hourly averages can be estimated with lower uncertainty. Going farther from the active vegetation, on the top of a mountain, seashore or island, in a desert or a poorly vegetated region, the concentration fluctuation is also lower than at a mid-continental site in the temperate zone. It means the atmospheric carbon dioxide models face a complex spatial-temporal structure of measurement uncertainty when incorporating data from tall towers applying sequential sampling. The uncertainty of the hourly data available in the databases may be magnitudes higher

in certain regions and times than that of the continuous measurements.

## 4 Conclusions

Accuracy of the atmospheric carbon dioxide budget calculation, source/sink allocations essentially depends on the accuracy of the measurements performed at the monitoring stations. Uncertainties derived from scale transfer, scale inconsistency, scale drift or instrument noise may be assessed (Vermeulen, 2016), and reduced with careful work, measurement

intercomparisons and applying high-quality instrumentation. Our analysis has shown that the uncertainty derived from the non-continuous sampling at the tall tower sites may be significantly higher than the other terms of the measurement uncertainty.

There is a good reason to suppose that the results presented in this paper are site-specific, and the uncertainties of the hourly average values in the databases vary in both time and space. Handling this variable uncertainty in the models, and assessing

its consequences on the results may be challenging. The data users should be aware of the spatially and temporally variable uncertainty of the measurement data they use. Metadata on sampling frequency and integration time, as well as more uncertainty studies, may help their work.

Because of the presumable site-specificity of the results, no general recommendation can be given for the measurement strategy. The trade-off between the high-frequency sampling to follow the atmospheric changes precisely and the long

integration time to get proper flushing and to reduce the instrument noise must be found locally knowing the local environmental conditions and the characteristics of the given monitoring system.

## Data availability

Data used in this study are available from the corresponding author upon request



**Author contribution**

LH initiated the project, performed most of the calculations, and wrote the manuscript. EP suggested and developed the mathematical methods for the study, and performed part of the calculations.

**Competing interest**

The authors declare that they have no conflict of interest.

**Acknowledgment**

The study was supported by the Hungarian National Research, Development and Innovation Office (project no.: K129118).

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
