# Peer review of "Uncertainty of the hourly average concentration values derived from non-continuous measurements"

_Atmospheric Measurement Techniques, 2020_

## Referee Comment (RC1) · Anonymous Referee #1 · 25 Nov 2020

General: The manuscript deals with the uncertainties of hourly averaged concentrations derived from non-continuous measurements as often applied at tall tower stations to optimize instrument investments, i.e. one instrument for several intakes and corresponding switching between different intake heights. This leads to continuous records only for limited periods within an hour and shifted in time for different intakes. The product build by the length of the flushing and sampling period times the frequency of the same intake reading depends on the number of intakes to be sampled within an hour. The manuscript clearly documents, which is to be expected, that the higher (lower) the frequency of one intake reading is (higher rate of switching between intakes) the lower (higher) deviations from the true hourly average (continuous records) and the lower

(higher) the uncertainties become. Furthermore, they investigated linear and cubic spline fitting averaging methods and compared these with a simple arithmetic mean. This study is important for modellers who would like to use them in different model approaches and thereby to have detailed information about uncertainties of hourly averages associated with non-continuous sampling. Despite this investigation is most-probably site-specific, as pointed out by the authors, it is worthwhile information how to deal with this issue at other stations. The manuscript is very nicely written with detailed information how the method works and how it is used and applied to data of a Hungarian station that exhibits very strong seasonalities and short-term fluctuations mainly due to photosynthesis/respiration processes. The figures and their legends are clear and concise. It was easy to read the manuscript and I would like to congratulate the authors. I have only a few rather minor comments and suggestions. I suggest to publish it once these comments have been taken into consideration. Minor points: Line 90: 35-45 s: Why does it take so long to reach the equilibrium values within 0.1 $\mu$mol mol-1 when the transfer time is less than 10 seconds and the flow 220 ml/min. The cell volume of the Picarro instrument in use (2301) is 33 ml and its regulated pressure I guess is at 140 Torr. Therefore, I would expect a rather rapid equilibration within a few seconds (e-folding time is 1.66 seconds equal to (33ml*140Torr/760Torr)/220ml/min*60s/min).

Line 104: I do not understand the values in parenthesis, please comment on them.

Line 155: I would rewrite this sentence to: At such a resolution the available data are insufficient in number for reliably estimate the scale parameter of the Cauchy distribution.

Line 157ff: The results are rather qualitative than quantitative but ...

Line 160ff: Do you have the data available also for the different interpolations (linear, cubic spline)? If yes, then add this information already here. Figure 3: Legend, change . . .the true value for a typical summer morning hour . . .

Line 198: Delete starting a new paragraph

Line 201: change the sentence to: The higher the sampling frequency the better the arithmetic mean mirrors the concentration course and the lower the uncertainty of the estimated hourly average becomes.
* * *

---

## Referee Comment (RC2) · Anonymous Referee #2 · 17 Dec 2020

Review of the manuscript: "Uncertainty of the hourly average concentration values derived from non-continuous measurements" by László Haszpra and ErnÅŚ Prácser (AMT)

The manuscript presents a short analysis of CO2 measurements at Hegyhatsall tall tower in Hungary, focusing on the calculation of the hourly aggregation errors made from incomplete measurement series. Indeed CO2 measurement tall towers generally only use one analyzer to measure concentrations at several (3-5) levels. As a result, the measurement period must be distributed between these different levels, and the hourly averages provided to international databases are calculated with an uncertainty

related to the representativeness of the measurement intervals, compared to the entire hour. The paper describes the methodology of this analysis very clearly, with very clear figures. I propose few minor corrections below, but I would like to see two important points developed before publication:

- the first point concerns the possibility of generalizing these results to other sites, by determining a relationship between the hourly aggregation uncertainty and the variability of the CO2 signal over the periods available.

- the second point concerns the discussion of the importance of hourly aggregation uncertainties for atmospheric inversions. It is essential to discuss more about random errors vs systematic errors, and to remember that atmospheric inversions currently only use measurements made when the atmosphere is well mixed, and therefore with minimal aggregation errors (according to your analysis).

Few comments:

Line 43: "which adds to the common instrument noise and scale uncertainty": I think there should be a clear distinction between random and systematic errors. Please specify that the latter are more critical in the context of atmospheric inversions, and those discussed in the paper are random errors.

Line 106-112: "we selected only those periods when the measurement elevation (82 m above the ground) was well within the planetary boundary layer" : I do not see the point of excluding those periods from the analysis. I would recommend to keep them and provide the results as a separate dataset

Figure 4: please clarify the period considered in this figure Line 180: "the shorter the sampling period the lower the uncertainty of the calculated hourly averages": I would rephrase this sentence to make it clear that the more injections of short duration, the lower the uncertainty.

Line 221-223: "It should be emphasized that the numerical results presented here may

AMTD
be highly site-specific..." : The question that the paper does not answer is whether the uncertainty is site dependent or a function of signal variability. Would it be possible to establish a relationship between the estimated uncertainty and the variability of the observed signal? With such a relation would it be possible to generalize the estimation of uncertainties?

Line 235: "Our analysis has shown that the uncertainty derived from the noncontinuous sampling at the tall tower sites may be significantly higher than the other terms of the measurement uncertainty": It seems to me that this conclusion deserves to be weighed. First of all, it is necessary to differentiate between random errors, such as the one discussed in this paper, from systematic errors such as those related to calibration scales. The latter is clearly more detrimental to the calculations of CO2 fluxes by the inverse methods. In addition, it should also be noted that most of atmospheric inversions only use data from tall towers during the afternoon, due to their difficulty in correctly reproducing the atmospheric dynamic the rest of the time. As a result, the data used in inversions correspond to those where the uncertainty of hourly aggregation is the lowest as shown by your analysis.

Line 242: "Metadata on sampling frequency and integration time, as well as more uncertainty studies, may help their work": Fully agree. One's could also considered to provide users with minute average concentrations rather than hourly average.

---

## Author Comment (AC1) · 31 Jan 2021

**Authors' response to Reviewer#1**

First of all, we would like to thank the Reviewer for his/her positive evaluation of our manuscript and would like to thank him/her for the comments and suggestions, which help us to improve the manuscript. Below you will find our detailed point-by-point response to the comments and suggestions:

Comment: *Line 90: 35-45 s: Why does it take so long to reach the equilibrium values within 0.1 μmol mol$^{-1}$ when the transfer time is less than 10 seconds and the flow 220 ml/min. The cell volume of the Picarro instrument in use (2301) is 33 ml and its regulated pressure I guess is at 140 Torr. Therefore, I would expect a rather rapid equilibration within a few seconds (e-folding time is 1.66 seconds equal to (33ml\*140Torr/760Torr)/220ml/min\*60s/min).*

Response: The Reviewer is right, theoretically stable reading should be achieved within a few seconds after the sample enters the measuring cell. The technical specification of Picarro G2301 declares <3 s for 10-90% / 90-10 % rise/fall response time, which indicates a somewhat longer time for achieving stable reading. One of the problems with measuring the response time of the system is the 5 s resolution of the readings. It can immediately introduce a maximum of 5 s error in both the beginning and end times of the flushing. Keeping in mind the temporal resolution and this type of error, our experience show 5-20 s stabilization time up to 20 ppm concentration difference. In the case of higher concentration difference between Standard 1 and Standard 2 the stabilization time gradually increases presumably due to some sort of a memory effect in the system. In the original manuscript the worst case was presented to support the decision on the flushing time, which is safely enough even in the worst case. This idea was not clearly exposed in the manuscript. In the revised manuscript we add that at usual concentration changes the response time is significantly shorter than 35-45 s.

Comment: *Line 104: I do not understand the values in parenthesis, please comment on them.*

Response: In the case of 2, 3, and 4 intakes 100 s sampling period results in 18, 12, and 9 full cycles through the intakes, respectively. 100 s sampling time cannot be applied for 5 intakes if full cycles are requested within an hour. (It would resulted in 7.2 cycles.) Because of the 5 s temporal resolution of the measurement, the sampling period has to be also divisible by 5. The shortest sampling time >100 s satisfying the requirements is 120 s, which results in 6 full cycles through the intakes within an hour. That is why the sampling period for 5 intakes are mentioned separately, in parentheses.

Comment: *Line 155: I would rewrite this sentence to: At such a resolution the available data are insufficient in number for reliably estimate the scale parameter of the Cauchy distribution.*

Response: The sentence will be corrected in the revised manuscript following the Reviewer's suggestion.

Comment: *Line 157ff: The results are rather qualitative than quantitative but ...*

Response: The sentence will be corrected in the revised manuscript following the Reviewer's suggestion.

Comment: *Line 160ff: Do you have the data available also for the different interpolations (linear, cubic spline)? If yes, then add this information already here. Figure 3: Legend, change ...the true value for a typical summer morning hour...*

Response: The requested information is available in the Supplementary material, although it was not clearly stated in the text. The sentence in line 161 of the original manuscript will be completed as follows: For other sampling periods, numbers of intakes, and averaging methods, the data are presented in the Supplementary material. The legend of Fig. 3 will be corrected according to the Reviewer's suggestion.

Comment: *Line 198: Delete starting a new paragraph*

Response: Accepted for the revised version of the manuscript.

Comment: *Line 201: change the sentence to: The higher the sampling frequency the better the arithmetic mean mirrors the concentration course and the lower the uncertainty of the estimated hourly average becomes.*

Response: We will replace the sentence with the suggested one.

---

## Author Response (AR1)

**Authors' response to Reviewer#1**

First of all, we would like to thank the Reviewer for his/her positive evaluation of our manuscript and would like to thank him/her for the comments and suggestions, which have helped us to improve the manuscript. Below you will find our detailed point-by-point response to the comments and suggestions:

Comment 1: Line 90: 35-45 s: Why does it take so long to reach the equilibrium values within 0.1  $\mu$ mol mol-1 when the transfer time is less than 10 seconds and the flow 220 ml/min. The cell volume of the Picarro instrument in use (2301) is 33 ml and its regulated pressure I guess is at 140 Torr. Therefore, I would expect a rather rapid equilibration within a few seconds (e-folding time is 1.66 seconds equal to (33ml\*140Torr/760Torr)/220ml/min\*60s/min).

Response: The Reviewer is right, theoretically stable reading should be achieved within a few seconds after the sample enters the measuring cell. The technical specification of Picarro G2301 declares <3 s for 10-90% / 90-10 % rise/fall response time, which indicates a somewhat longer time for achieving stable reading. One of the problems with measuring the response time of the system is the 5 s resolution of the readings. It can immediately introduce a maximum of 5 s error in both the beginning and end times of the flushing. Keeping in mind the temporal resolution and this type of error, our experience shows 5-20 s stabilization time up to 20 ppm concentration difference. In the case of higher concentration difference between Standard 1 and Standard 2, the stabilization time gradually increases presumably due to some sort of a memory effect in the system. In the original manuscript, the worst case was presented to support the decision on the flushing time, which is safe enough even in the worst case. This idea was not clearly exposed in the manuscript. The modified section (**line 90-94**) says:

"Our experience with standard gases shows that the deviation of the concentration in the measuring cell from the true value falls below  $0.1 \ \mu mol \ mol^{-1}$  within 5-20 s if the concentration difference is below 20  $\ \mu mol \ mol^{-1}$ . More precise determination of the response time is not possible due to the coarse temporal resolution of the readings (5 s), which may introduce an error up to 10 s immediately. In the case of higher concentration difference, the response time increases reaching 35-45 s at 70  $\ \mu mol \ mol^{-1}$  difference, which may indicate some sort of a memory effect in the system."

Figure S1 showing the extreme case has been deleted from the Supplementary material not to mislead the reader.

Comment 2: Line 104: I do not understand the values in parenthesis, please comment on them.

Response: In the case of 2, 3, and 4 intakes 100 s sampling period results in 18, 12, and 9 full cycles through the intakes, respectively. 100 s sampling time cannot be applied for 5 intakes if full cycles are requested within an hour. (It would result in 7.2 cycles.) Because of the 5 s temporal resolution of the measurement, the sampling period has to be also divisible by 5. The shortest sampling time >100 s satisfying the requirements is 120 s, which results in 6 full cycles through the intakes within an hour. That is why the sampling period for 5 intakes are

mentioned separately, in parentheses. The section has been rephrased as follows (line 112-116):

"Taking into account the above experience, we selected the shortest sampling time for this study as 100 s including 60 s flushing and 40 s signal averaging for the cases when 2, 3, or 4 intakes were assumed. For 5 intakes the shortest sampling time was 120 s including 60 s averaging to provide an equal number of sampling periods at each intake in an hour taking into account the 5 s temporal resolution of the instrument readings."

Comment 3: Line 155: I would rewrite this sentence to: At such a resolution the available data are insufficient in number for reliably estimate the scale parameter of the Cauchy distribution.

Response: The sentence has been corrected in the revised manuscript following the Reviewer's suggestion. See **line 188-189**.

Comment 4: *Line 157ff: The results are rather qualitative than quantitative but ...*

Response: The sentence has been corrected in the revised manuscript following the Reviewer's suggestion. See **line 190-192**.

Comment 5: Line 160ff: Do you have the data available also for the different interpolations (linear, cubic spline)? If yes, then add this information already here. Figure 3: Legend, change ...the true value for a typical summer morning hour...

Response: The requested information is available in the Supplementary material, although it was not clearly stated in the text. The sentence has been completed as follows (**line 193-194**):

"For other sampling periods, numbers of intakes, and averaging methods, the data are presented in the Supplementary material."

The legend of Fig. 3 has been corrected according to the Reviewer's suggestion (see **line 170**).

Comment 6: Line 198: Delete starting a new paragraph

Response: Accepted for the revised version of the manuscript (line 213).

Comment 7: Line 201: change the sentence to: The higher the sampling frequency the better the arithmetic mean mirrors the concentration course and the lower the uncertainty of the estimated hourly average becomes.

Response: The sentence has been replaced by the following one (line 216-217):

"The higher the sampling frequency the better the arithmetic mean mirrors the concentration course, and the lower the uncertainty of the estimated hourly average becomes."

**Authors' response to Reviewer#2**

First of all, we would like to thank the Reviewer for his/her positive evaluation of our manuscript and would like to thank him/her for the comments and suggestions, which have helped us to improve the manuscript. Below you will find our detailed point-by-point response to the comments and suggestions.

Comment 1: The first point concerns the possibility of generalizing these results to other sites, by determining a relationship between the hourly aggregation uncertainty and the variability of the  $CO_2$  signal over the periods available.

Response: The Reviewer is right, there is a certain relation between the signal variability and the uncertainty of the estimated hourly average concentrations. The signal variability within an hour consists of two terms: the variance caused by the trend and the variance of the detrended signal (VAR). The trend can be characterized by the slope of the regression line fitted to the raw data (REG). A few random tests proved that a bivariate (REG, VAR) regression model can estimate the absolute deviation of the calculated hourly averages from the true values with reasonable reliability. The coefficients of the model depend on the number of intakes and the sampling period. Under our conditions, the intra-hour trend is the dominant term, and it might also be true for other sites. The reason is fairly obvious: in the case of a significant trend within the hour, the start time of the first measurement period significantly determines the calculated hourly average. Depending on the start time, the highest/lowest values at the beginning/end of the hour are missed from the averaging. Unfortunately, at the operative tall-tower sites neither the intra-hour trend nor the variance of the detrended data series can be calculated because the measurement is non-continuous. Therefore, a regression model like mentioned above cannot help the generalization of our results. Where continuous measurements are available, our calculations can be repeated locally. Unfortunately, high-resolution data are not available in the databases, therefore, we could not perform any test calculations of this kind. In the revised manuscript, we mention the importance of the intra-hour trend governing the uncertainty of the estimated hourly average concentration values (line 238-243), and give information on the typical diurnal variation of the concentration at our monitoring site (Figure 7 in the revised manuscript). It might give a clue for the station operators to guess whether the uncertainty might be higher or lower at their sites.

"Under our conditions, the intra-hour trend is the dominant term determining the uncertainty of the hourly average concentrations, and it might also be true for other sites. The reason is fairly obvious: in the case of a significant trend within the hour, the start time of the first measurement period significantly determines the calculated hourly average. Depending on the start time, the highest/lowest values at the beginning/end of the hour are missed from the averaging. To give an impression on the temporal variation at Hegyhátsál tall tower site Figure 7 shows the diurnal variation of the concentration for July when the diurnal amplitude is the highest."

Comment 2: The second point concerns the discussion of the importance of hourly aggregation uncertainties for atmospheric inversions. It is essential to discuss more about random errors vs systematic errors, and to remember that atmospheric inversions currently

only use measurements made when the atmosphere is well mixed, and therefore with minimal aggregation errors (according to your analysis).

Response: In the revised manuscript, we distinguish between systematic error, which is not discussed in the study, and may cause bias in the model results, and random errors caused – among others - by the non-continuous sampling, and may increase the uncertainty of the model results (**line 38-43**).

"Both the systematic and random errors of the measurements decrease the reliability of the results of the atmospheric models. The systematic errors as the scale bias may distort the source/sink distributions calculated by the models, they may result false emission values, while the random errors of the measurements increase the uncertainty of the calculated values. Instrument noise, scale instability, and other processes may cause random errors in the measurements. In this paper, we focus on the random error caused by the non-continuous sampling of the continuous concentration signal, which increases the uncertainty of the calculated hourly average concentrations."

We, the authors of the paper, are experimentalists having limited information on the ongoing model developments; therefore, we should be careful with our statements. Traditionally, the atmospheric inverse models used only the early afternoon measurements of the continental sites because the atmosphere is the best mixed in these hours, and the spatial representativeness of the measurements is the highest under continental conditions. It meant that >80 % of the measurements were useless for the models. The progress in the representation of atmospheric dynamics in the models may make it possible to use data from a wider time-window, making more measurement data useful. However, the wider time-window also means that data with higher uncertainty also penetrate the model. We think, the uncertainties of these data, and their temporal variations are important background information for the modelers in the evaluation of the model results. We have added a paragraph to Conclusions (**line 268-272**) about this topic:

"In the case of continental monitoring sites, the present-day atmospheric inversion models typically use only the early afternoon measurements, the uncertainty of which is the lowest. However, it also means that ~80 % of the measurements are not used. The progress in the representation of atmospheric dynamics in the models may make it possible to use data from a wider time-window (e.g. from late morning till evening), making more measurement data useful. However, the wider time-window also means that data with higher uncertainty also penetrate the model."

Comment 3: Line 43: "which adds to the common instrument noise and scale uncertainty": I think there should be a clear distinction between random and systematic errors. Please specify that the latter are more critical in the context of atmospheric inversions, and those discussed in the paper are random errors.

Response: Agreeing with the Reviewer, we have added a paragraph to the revised manuscript making a clear distinction between random and systematic errors, and emphasizing their different effects on the modeling results (**line 38-43**):

"Both the systematic and random errors of the measurements decrease the reliability of the results of the atmospheric models. The systematic errors as the scale bias may distort the

source/sink distributions calculated by the models, they may result false emission values, while the random errors of the measurements increase the uncertainty of the calculated values. Instrument noise, scale instability, and other processes may cause random errors in the measurements. In this paper, we focus on the random error caused by the non-continuous sampling of the continuous concentration signal, which increases the uncertainty of the calculated hourly average concentrations."

Comment 4: Line 106-112: "we selected only those periods when the measurement elevation (82 m above the ground) was well within the planetary boundary layer" : I do not see the point of excluding those periods from the analysis. I would recommend to keep them and provide the results as a separate dataset.

Response: The top of the planetary boundary layer (PBL) drops below the measurement elevation almost exclusively during nights. The variability of the concentration in and above the planetary boundary layer is quite different. Mixing of the uncertainty values for these quite different regimes would lead to hardly interpretable results. They may not be characteristic for either those cases when the measurements represent the conditions in the PBL or those cases when they represent the conditions in the nighttime residual layer/free troposphere. In the study, the PBL height data from ERA5 reanalysis dataset were used, which also have their own uncertainty. To be sure that the measurements certainly represent the boundary layer conditions we set a lower limit for the PBL height (120 m) safely above the measurement elevation (82 m). For clarity we have completed the paragraph as follows (**line 117-121**):

"The dynamic processes in the planetary boundary layer (PBL) significantly differ from those above that. Mixing of the cases when the PBL is sampled, and when the air sample is taken from the free troposphere or from the nighttime residual layer above that would lead to a hardly interpretable result. Therefore, in the present study, we selected only those periods when the measurement elevation (82 m above the ground) was well within the planetary boundary layer, the depth of the PBL was at least 120 m at both the beginning and end of the hour considered."

Comment 5: Figure 4: please clarify the period considered in this figure Line 180: "the shorter the sampling period the lower the uncertainty of the calculated hourly averages": I would rephrase this sentence to make it clear that the more injections of short duration, the lower the uncertainty.

Response: Thank you for calling our attention to missing a piece of information from Figure 4. The figure shows the median and the 90-percentile values of the absolute deviation from the true hourly averages based on all data (whole year). The sentence in Line 180 of the original manuscript has been rephrased in the revised version (**line 166-168**):

"Figure 4 shows the  $\gamma$  values (equivalent with the half of the interquartile range or the median of the absolute deviations) and the 90-percentile values of the absolute deviations for the different number of intakes as a function of the sampling time and averaging method (arithmetic average, linear interpolation, cubic spline interpolation) based on the whole dataset." Comment 6: Line 221-223: "It should be emphasized that the numerical results presented here may be highly site-specific..." The question that the paper does not answer is whether the uncertainty is site dependent or a function of signal variability. Would it be possible to establish a relationship between the estimated uncertainty and the variability of the observed signal? With such a relation would it be possible to generalize the estimation of uncertainties?

Response: While the uncertainty directly depends on the signal variability, the signal variability is basically site-dependent. It depends on the geographical environment of the monitoring site (seashore, mountain top, low elevation mid-continental site, etc.), climate conditions, and the elevation of the measurements above the ground. The land-cover may also influence the signal variability through local turbulence. As it is discussed in detail in Response to Comment 1, a few random tests proved that a bivariate regression model based on the intra-hour trend and variance of the detrended data can estimate the absolute deviation of the calculated hourly averages from the true values with reasonable reliability. Unfortunately, at the operative tall-tower sites neither the intra-hour trend nor the variance of the detrended because the measurement is non-continuous, the measurement elevations are sampled sequentially. We reformulated the paragraph in line 221-230 of the original manuscript and add **Figure 7** (**line 233-244**)

"It should be emphasized that the numerical results presented here may be highly sitespecific. They depend on the signal variability, which may depend on the height of the sampling elevation above the ground, on the geographical location and environment of the monitoring site, and partly on instrument setup. Sources and sinks of carbon dioxide are located at the surface, and the fluctuation generated by the surface processes gradually attenuates with height (Stull, 1999). The relative role of the high-frequency part of the spectra is reduced at higher elevations, and so the hourly averages can be estimated with lower uncertainty. Under our conditions, the intra-hour trend is the dominant term determining the uncertainty of the hourly average concentrations, and it might also be true for other sites. The reason is fairly obvious: in the case of a significant trend within the hour, the start time of the first measurement period significantly determines the calculated hourly average. Depending on the start time, the highest/lowest values at the beginning/end of the hour are missed from the averaging. To give an impression on the temporal variation at Hegyhátsál tall tower site Figure 7 shows the diurnal variation of the concentration for July when the diurnal amplitude is the highest. The uncertainty of the hourly averages presented in Fig. 5 is in synchrony with the diurnal variation of the rate of concentration change."

**See also the Response to Comment 1.**

Comment 7: Line 235: "Our analysis has shown that the uncertainty derived from the noncontinuous sampling at the tall tower sites may be significantly higher than the other terms of the measurement uncertainty": It seems to me that this conclusion deserves to be weighed. First of all, it is necessary to differentiate between random errors, such as the one discussed in this paper, from systematic errors such as those related to calibration scales. The latter is clearly more detrimental to the calculations of CO2 fluxes by the inverse methods. In addition, it should also be noted that most of atmospheric inversions only use data from tall towers during the afternoon, due to their difficulty in correctly reproducing the atmospheric

**dynamic the rest of the time. As a result, the data used in inversions correspond to those where the uncertainty of hourly aggregation is the lowest as shown by your analysis.**

Response: Here we would repeat our response to Comment 2. In the revised manuscript we have made a distinction between the systematic errors (not discussed in the paper) and the random errors among which the uncertainty caused by the non-continuous sampling is studied (**line 38-43** – see at Response to Comment 2). We believe that with the development of the atmospheric inverse models more measurement data could be used, and we would like to call the modelers' attention that data with higher uncertainty will appear as input what has to be taken into consideration when the model results are discussed. We add a paragraph to Conclusions about this topic (**line 268-273**):

"In the case of continental monitoring sites, the present-day atmospheric inversion models typically use only the early afternoon measurements, the uncertainty of which is the lowest. However, it also means that ~80 % of the measurements are not used. The progress in the representation of atmospheric dynamics in the models may make it possible to use data from a wider time-window (e.g. from late morning till evening), making more measurement data useful. However, the wider time-window also means that data with higher uncertainty also penetrate the model."

Comment 8: Line 242: "Metadata on sampling frequency and integration time, as well as more uncertainty studies, may help their work": Fully agree. One's could also considered to provide users with minute average concentrations rather than hourly average.

Response: We have added a sentence to the paragraph (**line 277-278**) suggesting the submission of minute data. It would allow the users to perform their own statistical evaluations on the uncertainty of the aggregated data.

"It might also be reasonable to store the data with high temporal resolution (e.g. minute values) in the public databases."

---

## Author Response (AR2)

Dear Editor,

Thank you very much for reviewing our manuscript. We have accepted all corrections you have requested, and modified the manuscript accordingly.

Unfortunately, the short deadline and the bureaucratic processing time of any purchasing here have not allowed us to get and study the ISO standard before the composition of our response. Reconstruction of the formulas you provided makes it very likely that the ISO standard is valid only for data series without trend, i.e. for the cases when any subset of the data is representative for the whole period of the averaging. It is not necessarily the case for real-life atmospheric measurements, especially during the morning and evening transition periods. As Figure 3 in our paper illustrates, the start time of the measurements may critically influence the deviation of the calculated hourly average from the true one, because the subset of the data is not necessarily representative for the period of averaging.

We show in our paper that the more frequently an intake is sampled (the sampling period is shorter) the lower the uncertainty of the calculated hourly averages, because the measurements follow the temporal variation of the atmospheric concentration more closely. The higher the frequency the more data points are lost during the flushing periods. According to the ISO formulas, with higher sampling frequency the uncertainty would be increasing, not decreasing, due to the fewer data points available. This phenomenon indicates that the ISO standard cannot be applied when the concentration shows trend-like changes within the averaging period.

Yours sincerely

László Haszpra